# Mitochondria reorganization upon proliferation arrest predicts individual yeast cell fate

Damien Laporte, Laëtitia Gouleme, Laure Jimenez, Ines Khemiri, Isabelle Sagot*

Centre National de la Recherche Scientifique, Université de Bordeaux - Institut de Biochimie et Génétique Cellulaires, Bordeaux, France

**Abstract** Most cells spend the majority of their life in a non-proliferating state. When proliferation cessation is irreversible, cells are senescent. By contrast, if the arrest is only temporary, cells are defined as quiescent. These cellular states are hardly distinguishable without triggering proliferation resumption, hampering thus the study of quiescent cells properties. Here we show that quiescent and senescent yeast cells are recognizable based on their mitochondrial network morphology. Indeed, while quiescent yeast cells display numerous small vesicular mitochondria, senescent cells exhibit few globular mitochondria. This allowed us to reconsider at the individual-cell level, properties previously attributed to quiescent cells using population-based approaches. We demonstrate that cell's propensity to enter quiescence is not influenced by replicative age, volume or density. Overall, our findings reveal that quiescent cells are not all identical but that their ability to survive is significantly improved when they exhibit the specific reorganization of several cellular machineries.

DOI: https://doi.org/10.7554/eLife.35685.001

*For correspondence: isabelle.sagot@ibgc.cnrs.fr

**Competing interests:** The authors declare that no competing interests exist.

## Introduction

All cells integrate a complex combination of environmental cues and intracellular signals before committing to a new round of cell division. If these composite indicators are unfavourable to proliferation, the cell has several potential fates. First, it may enter quiescence, a reversible, non-dividing cellular state. As such, quiescent cells not only have to remain viable over time, but also have to maintain their proliferation capacity. Alternatively, a cell may enter senescence, a state in which cells remain metabolically active but will never divide again (*Campisi and d'Adda di Fagagna, 2007*; *Blagosklonny, 2011*; *O'Farrell, 2011*; *Valcourt et al., 2012*; *Cheung and Rando, 2013*; *Salama et al., 2014*; *Terzi et al., 2016*). Ultimately, cells die, either accidentally or in a programmed manner (*Nomenclature Committee on Cell Death 2009 et al., 2009*; *Galluzzi et al., 2012*). Consequently, within both unicellular organism colonies and eukaryotic tissues, cell populations are highly heterogeneous and composed of dividing, quiescent, senescent and dead cells. This heterogeneity is further complicated by the fact that with time, cells age. Cellular aging is viewed as a combination of genetically encoded processes and an accumulation of irreversible macromolecular damages (*López-Otín et al., 2013*; *Kowald and Kirkwood, 2016*). In fact, two aging paradigms have been defined. The replicative age is the number of divisions a cell can potentially undergo before entering senescence (*Mortimer and Johnston, 1959*; *Hayflick and Moorhead, 1961*). The chronological age is the time a non-proliferating cell remains viable and capable of giving rise to a progeny (*Fabrizio and Longo, 2003*), quiescent cells being by definition the archetype of chronologically aging cells.

To decipher the complex interplay between cell proliferation, quiescence, senescence and cell death, and understand the impact of cellular aging, each cell category has to be identified and

eventually tracked in space and possibly in time. This is a formidable technical challenge, particularly because quiescent cells can currently only be identified *a posteriori,* after experimentally testing their capacity to re-proliferate. Therefore, there is a crucial need for criteria recognizable in living cells that robustly correlate with the fate of non-dividing cells at the individual-cell level.

*S. cerevisiae* has been a powerful model for studying cellular aging. In this eukaryote, a single environmental change can induce various individual responses, even in a clonal population (*Honigberg, 2016*). For example, when a yeast population exhausts one nutrient, it enters a so-called 'stationary phase' (*Gray et al., 2004*). This population is heterogeneous and composed of quiescent, senescent and dead cells, the proportion of which evolves with time and differs depending on the nature of the exhausted nutrient (*Davidson et al., 2011*; *Klosinska et al., 2011*; *Werner-Washburne et al., 2012*; *Laporte et al., 2017*). Several laboratories have attempted to identify each cell category according to differences in their physical properties. The Werner-Washburne laboratory has pioneered these studies utilizing a density gradient that separates the stationary phase population into two sub-fractions (*Allen et al., 2006*). This study led to a Boolean concept in which only dense small daughter cells were considered as *bona fide* quiescent cells, while the light fraction, called non-quiescent, supposedly contained senescent and dead mother cells. A corollary to this dichotomy is that replicative age strongly impacts the cell's ability to face chronological age. Yet, as acknowledged later by the authors, this model is over-simplistic, as both sub-populations are highly heterogeneous and do contain quiescent cells (*Aragon et al., 2008*; *Davidson et al., 2011*; *Werner-Washburne et al., 2012*). More recently, centrifugal elutriation was used to separate cells of a stationary phase culture according to their volume. The authors showed that a sub-population of very small daughter cells (2–4 µm in diameter) contains mostly senescent or dead cells, challenging thus the 'density model' (*Svenkrtova et al., 2016*). These discrepancies highlight the limitations of cell population sub-fractionation techniques, their strongest caveat being to assign to sub-populations properties that define individual cells.

Mitochondria play key roles in cellular aging (*Sun et al., 2016*; *Kauppila et al., 2017*; *Sebastián et al., 2017*). These organelles are metabolic centers involved in multiple essential cellular processes, including the production of ATP by oxidative phosphorylation (OXPHOS) (*Nunnari and Suomalainen, 2012*). In *S. cerevisiae*, mitochondrial respiration dysfunctions severely affect cell's ability to face both replicative and chronological aging, (*Fabrizio and Longo, 2003*; *Martinez et al., 2004*; *Trancíková et al., 2004*; *Bonawitz et al., 2006*; *Aerts et al., 2009*; *Ocampo et al., 2012*; *Beach et al., 2015*). Importantly, mitochondrial morphology defects have been observed in replicatively-old yeast mother cells (*Scheckhuber et al., 2007*; *Veatch et al., 2009*; *McFaline-Figueroa et al., 2011*; *Hughes and Gottschling, 2012*; *Fehrmann et al., 2013*). In proliferating yeast, mitochondria form a dynamic tubular meshwork (*Friedman and Nunnari, 2014*) but intriguingly, in the early 1960's, Yotsuyanagi reported that upon proliferation cessation, mitochondria reorganize into peripherally-localized vesicles (*Yotsuyanagi, 1962*), a rearrangement also observed more recently (*Stevens, 1981*; *Dürr et al., 2006*; *Sagot et al., 2006*), but never investigated.

Here we show that following carbon source exhaustion, proliferation cessation is accompanied by a drastic reorganization of the mitochondrial network. We demonstrate that in cells entering quiescence, mitochondrial tubules progressively fragment into numerous small cortical vesicles. In contrast, cells that will enter senescence display few large globular mitochondria that form upon the diauxic shift. These observations enable us to distinguish quiescent and senescent cells within a heterogeneous stationary phase population. We thus re-examined, at the single-cell level, some controversial properties assigned to quiescent cells using population-based approaches. We show that neither cell density nor cell volume could discriminate cell fate. Importantly, we establish that replicative age does not impinge on a cell's ability to face chronological aging, at least for the first 15 divisions. Finally, we analyze the significance of actin cytoskeleton reorganization into Actin Bodies (AB) and proteasome re-localization from the nucleus into proteasome storage granules (PSG) previously observed in non-dividing cells (*Sagot et al., 2006*; *Laporte et al., 2008*). Overall, our findings reveal that not all quiescent cells are identical, yet quiescence establishment predominantly ensues upon the reorganization of specific cytoplasmic elements.

## Results

### Mitochondrial network reorganization upon proliferation cessation

Mitochondrial network morphology was analyzed in wild-type (WT) cells expressing the mitochondrial matrix protein Ilv3 fused to RFP (*Jimenez et al., 2014*), the expression of which having no effect on cell proliferation (*Figure 1—figure supplement 1A*). Cells proliferating in glucose-rich liquid medium (YPD) displayed a dynamic branched tubular mitochondrial network, as expected from previous studies. Interestingly, we observed that upon glucose exhaustion, mitochondria tubules progressively fragmented. With time, these fragments appeared shorter and shorter and after 7 days, the population was composed of ~85% of cells displaying small and immobile cortical mitochondrial vesicles (>50 per cell) that may be an ultimate form of tubule fragmentation. In addition, we found ~10% of cells exhibiting 2 to 4 large globular mitochondria and ~5% of cells in which we could not detected any Ilv3-RFP signal (*Figure 1A*). As cultures aged, the percentage of non-fluorescent cells increased, the percentage of cells with vesicular mitochondria decreased and the percentage of cells with globular mitochondria remained stable (*Figure 1—figure supplement 1B*). Importantly, as in WT cells, after 7 days of growth in YPD, vesicular and globular mitochondria were observed in *dnm1Δ*, *fis1Δ*, *caf4Δ*, *mdv1Δ* and *atg32Δ* cells (*Figure 1B*). Together, these data demonstrate that upon proliferation cessation following glucose exhaustion, the mitochondrial network is drastically reshaped and thus, independently of both the mitochondria fission machinery (*Labbé et al., 2014*; *Friedman and Nunnari, 2014*) and selective mitophagy (*Kanki et al., 2009*; *Okamoto et al., 2009*).

At the individual cell level, mitochondrial DNA (mtDNA) was detected in 89 ± 1% and 96% ± 1% of 7 days old WT cells with globular and vesicular mitochondria respectively (*Figure 1C*). Moreover, the mtDNA binding protein Abf2-GFP (*Diffley and Stillman, 1991*), the MICOS complex (*Pfanner et al., 2014*), the Tim/Tom transporters (*Harbauer et al., 2014*) and other components belonging to the mitochondrial matrix, the inner and the outer membranes were detected in both cells types (*Figure 1D* and *Figure 1—figure supplement 1C–E*). However, while vesicular mitochondria contained both respiratory chain enzymes and ATP synthase subunits, some of OXPHOS proteins were not detected in 7-days-old cells with globular mitochondria (*Figure 1E* and *Figure 1—figure supplement 1C–E*). These led us to speculate that cells with globular mitochondria may be respiration deficient. To test this idea, we took advantage of the observation that the formation of cells with globular mitochondria could be triggered by switching proliferating cells from fermentation to respiration, their percentage reaching a plateau at 8.7 ± 0.1%, 4 hr after the carbon source exchange (*Figure 1F*). The formation of cells with globular mitochondria was induced as above and 4 hr after the medium switch, cells were pulse-stained with FITC-conjugated concanavalin A (ConA-FITC), a lectin that binds the yeast cell wall. Cells were then released in YPGE or in YPD as a control. While the percentage of ConA-FITC positive cells with globular mitochondria rapidly decreased in YPD, it remained constant in YPGE, even 15 hr after the pulse (*Figure 1G*). In addition, within ConA-FITC positive cells with globular mitochondria, the mother/daughter cells ratio remained constant (*Figure 1H*, right), confirming that these cells did not proliferate in YPGE. By comparison, the ratio increased for cells with tubular mitochondria (*Figure 1H*, left). Taken together, our data demonstrate that cells with globular mitochondria are respiratory deficient. Accordingly, no cells with globular mitochondria were observed in cultures grown in rich medium containing respiratory carbon sources (glycerol/ethanol or lactate, *Figure 1—figure supplement 1F*).

### Mitochondrial network reorganization can predict cell fate

We next investigated 7 days-old cell's ability to re-proliferate depending on the morphology of their mitochondrial network. Methylene blue staining revealed that unlike non-fluorescent cells, cells with vesicular or globular mitochondria were metabolically active (*Figure 2A*, *Painting and Kirsop, 1990*). To test cells ability to re-enter proliferation at the individual cell level, WT cells expressing Ilv3-RFP were grown for 7 days then transferred onto a microscope agarose pad. The pad contained glucose-rich medium in order to refeed the cells and thus induce cell re-proliferation. Individual cells were tracked and imaged every hour, up to 6 hr, a time after which extensive cell proliferation on the pad prevented us from unambiguously following individual cells. A cell was considered as exiting quiescence when a new bud emerged.

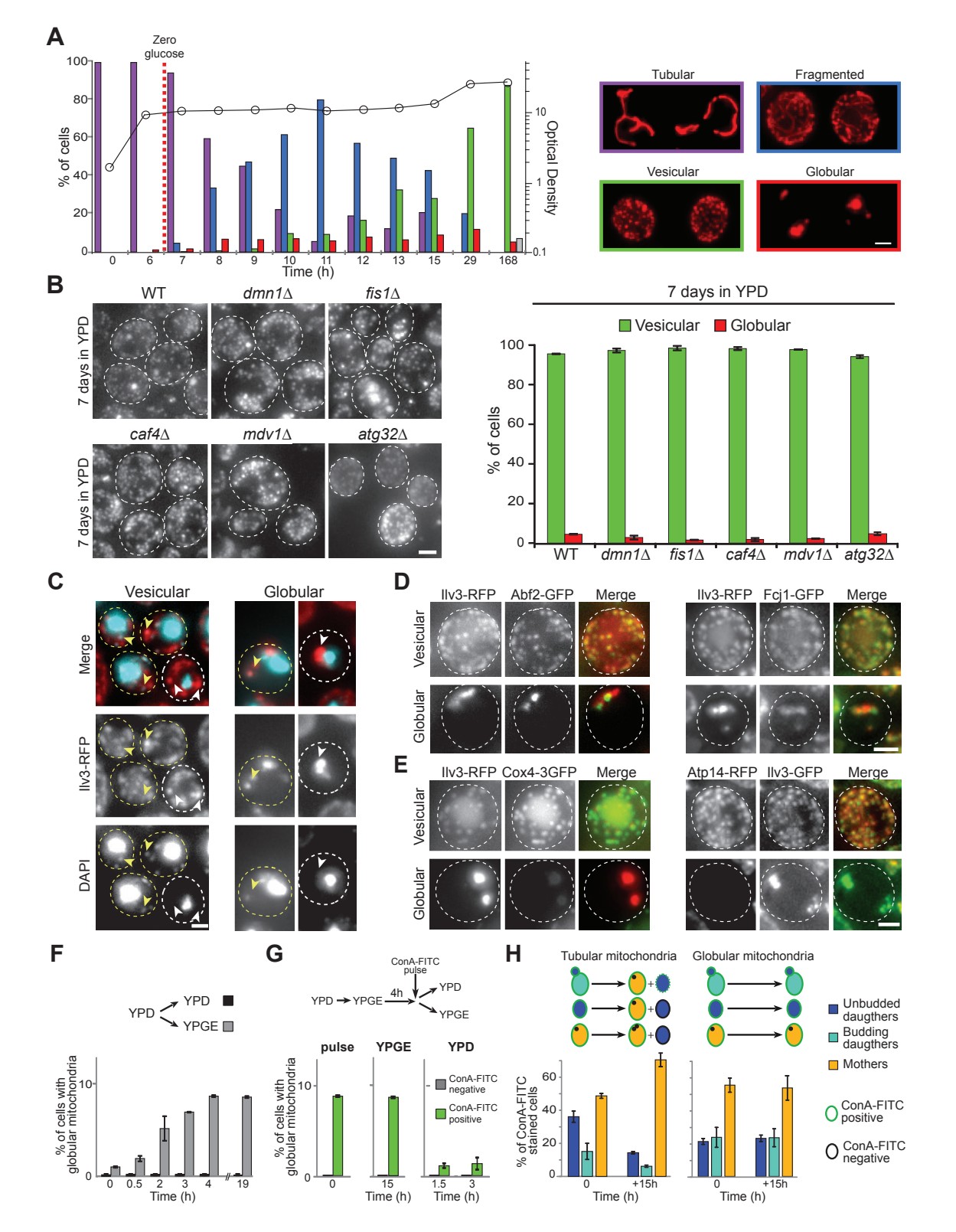

**Figure 1.** Mitochondrial network reorganization in non-proliferating cells. (**A**) Mitochondrial network organization in WT cells expressing Ilv3-RFP as a function of time in YPD (OD$_{600nm}$: circles). Cells with a tubular (violet), a fragmented (blue), a vesicular (green), a globular (red) mitochondrial network or no Ilv3-RFP signal (grey) were scored. Representative cells of each category are shown; n > 500, N = 2. (**B**) WT, *dmn1Δ*, *fis1Δ*, *caf4Δ*, *mdv1Δ*, and *atg32Δ* cells expressing Ilv3-RFP were grown 7 d in YPD. Histograms display the percentage of Ilv3-RFP positive cells with a vesicular (green) or a globular (red)

*Figure 1 continued on next page*

*Figure 1 continued*

mitochondrial network; n > 200, N = 2. (**C**) WT cells expressing Ilv3-RFP were grown 7 d in YPD and stained with DAPI. Yellow and white arrows point to stained or unstained mitochondria respectively. (**D**) WT cells grown 7 d in YPD co-expressing Ilv3-RFP and the mitochondrial DNA-binding protein Abf2-GFP or the MICOS complex protein Fcj1-GFP. (**E**) WT cells grown 7 d in YPD co-expressing Ilv3-RFP and the respiratory chain component Cox4-GFP or co-expressing Ilv3-GFP and the ATP synthase subunit Atp14-RFP. (**F**) WT cells expressing Ilv3-RFP were grown in YPD to $OD_{600nm}$ ~ 1. Cells were then shifted to YPD (black) or YPGE (grey). Cells with globular mitochondria were scored; n > 300, N = 2. (**G**) WT cells expressing Ilv3-RFP were grown in YPD to $OD_{600nm}$ ~ 1. Cells were then shifted to YPGE. After 4 hr, cells were pulse stained with ConA-FITC and shifted to YPD or YPGE. The percentage of cells with globular mitochondria was scored within both ConA-FITC stained (green) and unstained (grey) cell populations; n > 250, N = 2. (**H**) Cells from (**G**) were stained with calcofluor white concomitantly to the ConA-FITC pulse or 15 hr after the transfer to YPGE. The percentage of ConA-FITC positive mother (orange), unbudded daughter (dark blue) and budding daughter (light blue) cells were scored; n > 250, N = 2. Histograms show means, error bars are SD, bar is 2 μm.

DOI: https://doi.org/10.7554/eLife.35685.002

The following source data and figure supplements are available for figure 1:

**Source data 1.** Cell category scoring for each replicated experiment in *Figure 1* panel A, B and F to H.

DOI: https://doi.org/10.7554/eLife.35685.005

**Figure supplement 1.** Mitochondrial network organization in WT cells experiencing an extended period of non-proliferation or upon starvation for various nutrients and localization of various mitochondrial proteins in non-proliferating cells.

DOI: https://doi.org/10.7554/eLife.35685.003

**Figure supplement 1—source data 1.** Cell category scoring for each replicated experiment in *Figure 1—figure supplement 1* panel A, B and F.

DOI: https://doi.org/10.7554/eLife.35685.004

Using this individual cell-tracking strategy, we found that more than 90% of the 7-days-old cells that displayed vesicular mitochondria emitted a new bud within 6 hr after re-feeding (*Figure 2B*). As such, they were considered as quiescent (Chi test value $<10^{-26}$). Interestingly, upon quiescence exit, mitochondria vesicles rapidly fused and reformed short tubules before bud emergence (*Figure 2—figure supplement 1A*). Of note, in some experiments, we noticed that the cells could display both a vesicular and a vacuolar Ilv3-RFP staining, the latter resulting either from mitophagy or an increased Ilv3-RFP degradation in the vacuole. These cells had similar re-proliferation capacities than cells with an Ilv3-RFP signal exclusively localized into cortical vesicles (*Figure 2—figure supplement 1B*). Thus, both cell types were considered as a unique cell category.

We then wonder if increasing the time allowed to re-proliferate would increase the number of cells able to exit quiescence. To prevent pad overcrowding, highly diluted suspensions of 7 days old cells were deposited on YPD pads and cell re-proliferation was scored 12 hr after re-feeding. We found that 98 ± 2% of the cells with vesicular mitochondria were able to re-proliferate within 12 hr (*Figure 2C*). In conclusion, almost all cells with vesicular mitochondria were quiescent, although some cells re-entered the proliferation cycle faster than others.

By contrast,~90% of the 7-days-old cells displaying globular mitochondria were unable to re-enter proliferation, even after 12 hr on microscope pad (*Figure 2B–C*). These cells were thus considered as senescent (Chi test value $<10^{-140}$). To address the kinetic of entry into senescence, cells with globular mitochondria re-proliferation capacities were analyzed at various time points after their formation, that is after the diauxic shift. As shown in *Figure 2D*, the percentage of cells with globular mitochondria able to re-proliferate within 6 hr on a pad progressively decreased. After 7 days, less than 10% of these cells were able to re-proliferate, in agreement with *Figure 2B*. Thus, while cells with globular mitochondria were unable to proliferate under respiratory conditions whatever their age, the formation of globular mitochondria was not associated with an immediate loss re-proliferation capacity in glucose rich medium. Nevertheless, after 7 days, most of these cells appeared senescent. These data indicate that respiratory deficient cells with globular mitochondria enter senescence, but in a non-synchronous manner.

To study the link between globular mitochondria, respiration capacities and senescence, we first artificially increased the number of respiratory deficient cells using ethidium bromide (EB, (*Slonimski et al., 1968*) and *Figure 2E*). This treatment induced an increase in the percentage of cells with globular mitochondria (*Figure 2F*), more than 90% of which being senescent after 7 days (*Figure 2G*). In parallel, we observed that following carbon source exhaustion, rho zero cells rearranged their mitochondrial network into 5 to 20 round and swollen mitochondria (*Figure 2—figure supplement 1D*). After 14 days, these cells were alive but unable to re-proliferate on a microscope

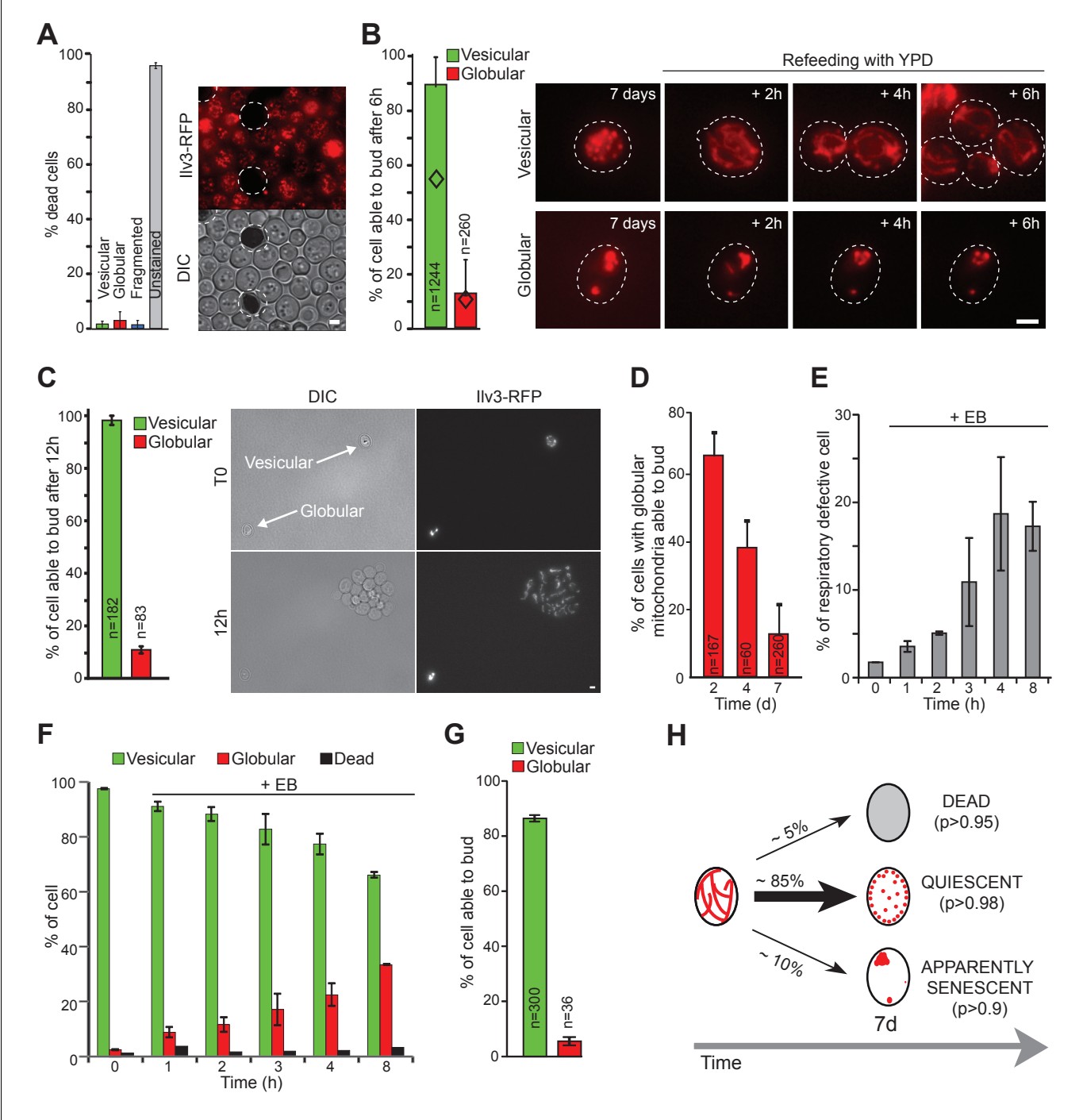

**Figure 2.** Mitochondrial network reorganization can predict cell fate. (**A**) WT cells expressing Ilv3-RFP were grown 7 d in YPD and stained with methylene blue to reveal dead cells. Cells surrounded by a white dashed line are methylene blue positive. The histogram shows the percentage of dead cells, n > 500, N = 2. (**B**) WT cells expressing Ilv3-RFP were grown 7 d in YPD and re-fed onto a microscope agarose pad containing YPD. Individual cells were imaged every hour. The percentage of cells able to form a new bud within 6 hr is shown (N = 4). Diamonds represent the theoretical independence value. (**C**) Same as in (**B**) but cells were imaged for 12 hr. The percentage of cells able to form a bud within 12 hr is shown (N = 3). (**D**) WT cells expressing Ilv3-RFP were grown in YPD for the indicated time and then refed onto a YPD microscope pad. The percentage of cells with globular mitochondria able to form a bud within 6 hr was scored (N = 2). (**E–G**) WT cells expressing Ilv3-RFP were grown in YPD to $OD_{600nm} \sim 0.1$. Cells were then treated with 3.5 µM etidium bromide (EB). After incubation for the indicated time, cells were washed and cultured in YPD. After 5 d, cells were plated on YPD, then replicated on YPGE. (**E**) The percentage of respiratory deficient colonies were scored n > 200, N = 3. (**F**) After 7 d, the percentage of dead cells (methylene blue positive) or cells with vesicular (green) or globular (red) mitochondria were scored. n > 300, N = 2. (**G**) After 7

*Figure 2 continued on next page*

*Figure 2 continued*

d, cells incubated 8 hr with EB were individually tested for their ability to re-proliferate on YPD microscope pads. The percentage of cells able to form a bud within 6 hr was scored (N = 2). (H) Mitochondrial network organization after 7 days in YPD and individual cell fate; (p) indicates the probability for each cell category 12 hr after refeeding on YPD. Bars are 2 μm. Error bars are SD.

DOI: https://doi.org/10.7554/eLife.35685.006

The following source data and figure supplements are available for figure 2:

**Source data 1.** Cell category scoring for each replicated experiment in *Figure 2* panel A, B, C, D and E to G.

DOI: https://doi.org/10.7554/eLife.35685.009

**Figure supplement 1.** Mitochondrial network organization in rho zero cells, ERMES mutant and WT cells of several genetic backgrounds upon proliferation cessation and quiescence exit.

DOI: https://doi.org/10.7554/eLife.35685.007

**Figure supplement 1—source data 1.** Cell category scoring for each replicated experiment in *Figure 2—figure supplement 1* panel A, C, E, F and G.

DOI: https://doi.org/10.7554/eLife.35685.008

pad or to give rise to a progeny on plate (*Figure 2—figure supplement 1C–D*). As such, 14 days old rho zero cells can be considered as senescent. Finally, we studied the ERMES mutant *mmm1Δ*. Proliferating *mmm1Δ* cells are known to display enlarged mitochondria (*Dimmer et al., 2002*; *Hanekamp et al., 2002*). Accordingly, after 7 days of culture, *mmm1Δ* cells exhibited abnormally big mitochondria (*Figure 2—figure supplement 1E*). As expected, *mmm1Δ* cells were massively respiratory deficient (98 ± 4%). After 16 days, we found that most *mmm1Δ* cells appeared senescent (*Figure 2—figure supplement 1E*). Taken together these experiments indicate that the inability to metabolize respiratory carbon sources correlates with the inability to maintain quiescence.

Finally, we observed that the mitochondrial network morphology was also drastically modified following the exhaustion of other nutrients. As stated above, upon respiratory carbon source exhaustion (glycerol/ethanol and lactate), no cell with globular mitochondria were observed, the majority of the cells displaying a vesicular network and being quiescent. Phosphate depletion, just as carbon depletion, led to both globular and vesicular mitochondria formation and more than 90% of the cells were able to form a colony. By contrast, as shown previously, nitrogen exhaustion led to massive mitophagy (*Camougrand et al., 2008*) and rapidly caused cell death (*Figure 1—figure supplement 1F*).

In conclusion, the above data demonstrate that following carbon source exhaustion, the mitochondrial network organization correlates with individual cell fate: unstained cells are dead, cells with globular mitochondria appeared senescent and cells with vesicular mitochondria are quiescent (*Figure 2H*), these findings being reproducible in both the W303 and the CEN-PK strain backgrounds (*Figure 2—figure supplement 1F–G*). Hence, the mitochondrial network morphology can predict the fate of yeast cells facing chronological aging. We took advantage of this robust hallmark to revisit, at the individual cell level, some controversial quiescent cell properties that were uncovered at a population scale.

## Cell volume and the ability to face chronological age

Cell volume has been suggested to strongly influence non-proliferating cell survival. Yet while the Werner-Washburne laboratory found that only small daughter cells were quiescent (*Allen et al., 2006*; *Aragon et al., 2008*; *Werner-Washburne et al., 2012*; *Davidson et al., 2011*), the Pichova lab reported that a sub-population composed of very small cells (2–4 μm in diameter) contained mostly senescent or dead cells (*Svenkrtova et al., 2016*). To revisit these conflicting data, we measured the individual cell volume of WT cells expressing Ilv3-RFP grown in YPD, just after their deposition onto a YPD microscope pad. The volume of 7-days-old cells ranged from ~10 fL up to ~180 fL (*Figure 3A*). With time, the population median cell volume slightly decreased (from 43.3 fL at 7 days to 36 fL after 21 days; *Figure 3A*), a reduction probably resulting from cell desiccation, as cytoplasmic viscosity increases in non-proliferating *S. cerevisiae* (*Joyner et al., 2016*; *Munder et al., 2016*). For sake of simplicity, we split the population into small, medium and large cells, each sub-population corresponding to 1/3 of the total population (*Figure 3A*, red dash lines).

First, methylene blue staining revealed that dead cells (black bars) were evenly distributed among the population, whatever the population age, indicating that cell volume and cell death are uncorrelated (*Figure 3A*, Chi-test value >0.1). Second, we analyzed mitochondrial morphology and found

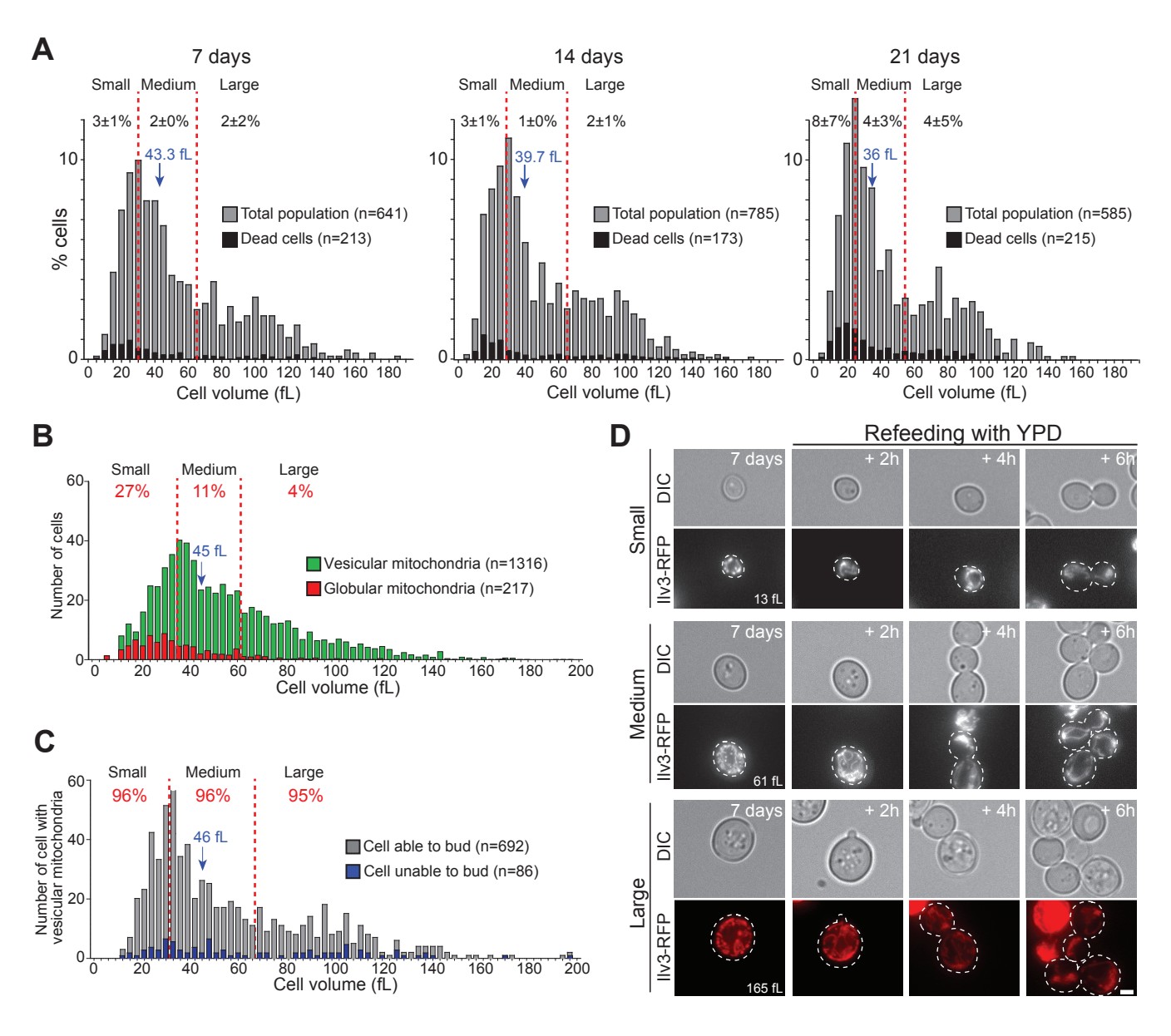

**Figure 3.** Cell volume and individual cell fate. (A) Cell volume distribution of WT cells grown for 7, 14, and 21 d in YPD. In A to C, red dashed lines define three sub-populations each representing 1/3rd of the total population. Dead cells (black bars) were identified using methylene blue staining (N = 3). Black numbers indicate the percentage of dead cells within each cell sub-population. (B) Cell volume distribution according to the mitochondrial organization (vesicular: green, globular: red, N = 3) of 7-d-old WT cells expressing Ilv3-RFP. Red numbers indicate the percentage of cells with globular mitochondria within each cell sub-population. (C) Cell volume distribution of 7-d-old WT cells expressing Ilv3-RFP with vesicular mitochondria according to cell re-proliferation capacity. Red numbers indicate the percentage of cells able to bud within 6 hr after refeeding onto a YPD microscope pad within each sub-population (for small cells, n = 235, medium cells n = 226 and large cells n = 231). In A to C, the population median cell volume is indicated in blue. (D) Image series of individual small, regular and large cells with vesicular mitochondria upon refeeding on a YPD microscope pad. Cell volumes before re-feeding are indicated. Bar is 2 μm.

DOI: https://doi.org/10.7554/eLife.35685.010

The following source data and figure supplements are available for figure 3:

**Source data 1.** Cell category scoring for each replicated experiment in *Figure 3* panel A, B, and C.
DOI: https://doi.org/10.7554/eLife.35685.013

**Figure supplement 1.** Cell viability and quiescence exit capacity after proliferation cessation in function of cell volume.
DOI: https://doi.org/10.7554/eLife.35685.011

**Figure supplement 1—source data 1.** Cell category scoring for each replicated experiment in *Figure 3—figure supplement 1* panel A and B.

*Figure 3 continued on next page*

*Figure 3 continued*

DOI: https://doi.org/10.7554/eLife.35685.012

that the 'small sub-population' was enriched in cells with globular mitochondria (*Figure 3B*). Thus, to some extent, we found more senescent cells among very small cells, in agreement with (*Svenkrtova et al., 2016*). Accordingly, when cells were micro-manipulated on plate to individually test their re-proliferation capacities (*Laporte et al., 2011*), small cells were less prone to form colonies than large cells (*Figure 3—figure supplement 1A*). Finally, when we considered exclusively cells with vesicular mitochondria, we found that cell's ability to re-proliferate was not influenced by the cell volume (*Figure 3C–D*). These results demonstrate that cells with vesicular mitochondria enter quiescence whatever their volume. In agreement, using a collection of cell-size mutants, we found that a cell's capacity to undergo chronological aging was unrelated to cell volume (*Figure 3—figure supplement 1B*).

## Replicative age does not impinge on chronological aging at a population level

After the study by Allen et al., it was taken for granted that only daughter cells were *bona fide* quiescent cells, in other words that a cell's replicative age impinges on its ability to go through chronological aging (*Allen et al., 2006*; *Aragon et al., 2008*; *Davidson et al., 2011*; *Werner-Washburne et al., 2012*; *Li et al., 2013*). We revisited these findings using our individual cell tracking approach. First, we found that 4 hr after the diauxic shift, mother and daughter cells have a similar propensity to form globular mitochondria (15 ± 6% and 11 ± 4% respectively (p-value=0.049)). Next, we observed that 7-days-old mother and daughter cells with vesicular mitochondria have a similar ability of exiting quiescence within 6 hr (*Figure 4A*). We then scrutinized mother cells and studied their re-proliferation capacity as a function of their replicative age that is the number of bud scars. As expected, the older the mother, the less frequent in the total population (*Figure 4B*). Interestingly, the percentage of cells with globular or vesicular mitochondria was not significantly different between mother cell replicative age categories (*Figure 4C*). Finally and importantly, we found that within mother cells with vesicular mitochondria, the replicative age did not influence cell re-proliferation capacities (*Figure 4D*). Notwithstanding, since cells with more than 15 bud scars are very rare in a yeast population (theoretically $1/2^{\text{number of division}}$), we could not study these very old mother cells with a statistical significance using our individual cell approach. Thus, cell's replicative age does not impinges on cell ability to face chronological aging, at least for cells that have undergone less than 15 divisions.

## Cell density and chronological aging

At a population level, quiescent cells were described as denser than non-quiescent cells (*Allen et al., 2006*). To revisit these data, 7 days-old WT cells expressing Ilv3-RFP were separated using a percoll density gradient (*Allen et al., 2006*). Although, the upper fraction was enriched in cells with globular mitochondria (21 ± 4%) and dead cells (11 ± 1%), it contained ~70% of cells with vesicular mitochondria (*Figure 4E*). By comparison, the lower fraction was composed of 96 ± 0% and 1 ± 0% of cells with vesicular or globular mitochondria respectively and 3 ± 0% of dead cells. To test the ability of individual cells to re-proliferate in function of their density, cells from a non-fractionated population were stained with ConA-AlexaFluor647, cells of the lower fraction were stained with ConA-FITC, and cells from the upper fraction left unstained. The three sub-populations were then mixed together, stained with calcofluor-white, and transferred onto the same YPD microscope pad. Using this strategy, we could compare the capacity of the different cell categories to exit quiescence under identical re-feeding conditions. We found that 96%, 96% and 95% of cells with vesicular mitochondria from the unfractionated population, the upper fraction and the lower fraction, respectively, were able to re-enter proliferation. These results demonstrate that cells with vesicular mitochondria can enter quiescence whatever their density. Moreover, in agreement with *Figure 4A*, among cells with vesicular mitochondria, mother cells were able to re-proliferate with the same efficiency than daughter cells whatever the fraction they belonged to (*Figure 4F*).

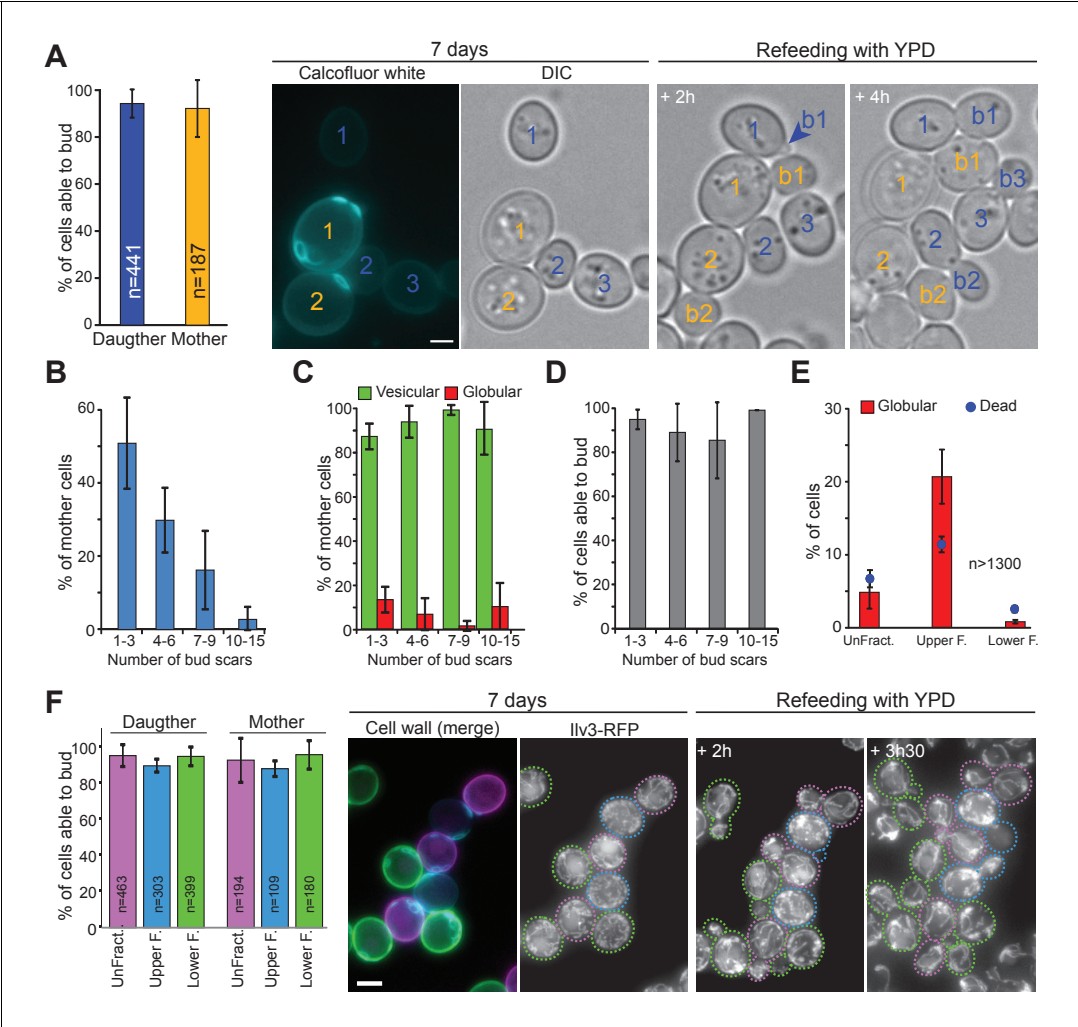

**Figure 4.** The capacity to enter quiescence does not correlate with a cell's replicative age or density. (A–D) WT cells expressing Ilv3-RFP were grown 7 d in YPD, stained with calcofluor white and refed onto a YPD microscope pad. (A) Percentage of mother or daughter cells with vesicular mitochondria able to form a bud within 6 hr after refeeding; n > 270, N = 3. Representative image series are shown. Number in yellow indicate mothers, in blue, daughters. (B) Distribution of the different mother cell categories according to the number of bud scars within the population of mother cells; n > 600, N = 3. (C) Percentage of cells with vesicular (green) or globular (red) mitochondrial network within each mother cell category; n > 400, N = 3. (D) Individual cell's ability to form a new bud within 6 hr after refeeding onto a YPD microscope pad according to the mother cell replicative age category; n > 400, N = 3. (E) and (F) WT cells expressing Ilv3-RFP were grown 7 d in YPD then centrifuged through a percoll density gradient. (E) The percentage of dead cells and cells with globular mitochondria were scored; n > 500, N = 2. (F) A non-fractionated population (unFract.) was stained with ConA-AlexaFluor647 (pink). Within the fractionated population, the lower fraction (lower F.) was stained with ConA-FITC (green). These two cell subpopulations were re-mixed with the upper fraction subpopulation (upper F.) that was left unstained (blue). Cells were then stained with calcofluor white and allowed to re-proliferate on a YPD microscope pad. The percentage of individual cells with vesicular mitochondria from each fraction that were able to bud within 6 hr after refeeding was scored (N = 3). A representative image series is shown. On the left a field merging the FITC (green), the AlexaFluor647 (pink) and the calcofluor white (blue) channels is shown. Cells are surrounded by dashed line with the color code indicated above. Bar are 2 μm. Error bars are SD.

DOI: https://doi.org/10.7554/eLife.35685.014

The following source data is available for figure 4:

**Source data 1.** Cell category scoring for each replicated experiment in *Figure 4* panel A, B to D, E and F.

DOI: https://doi.org/10.7554/eLife.35685.015

## Relationships between mitochondria organization, Actin Bodies and PSG

We have previously reported that upon carbon source exhaustion, the actin cytoskeleton reorganizes into Actin Bodies (AB) and the proteasome re-localizes from the nucleus into proteasome storage granules (PSG) (*Sagot et al., 2006*; *Laporte et al., 2008*; *Laporte et al., 2015*). Whether these rearrangements are specific of the quiescent state has been challenged (*Vasicova et al., 2015*), but never tested directly. To tackle this question, we used our individual cell tracking approach with cells expressing both Ilv3-RFP and either Abp1-3xGFP (AB) or Scl1-3xGFP (PSG). As expected from our previous studies, after 7 days in YPD, 85 ± 9% and 69 ± 1% of the cells displayed AB and PSG, respectively. More than 90% of the cells that displayed AB or PSG re-entered proliferation within 6 hr after re-feeding on a microscope pad (*Figure 5A*). Statistical analys indicated that AB or PSG formation and cell re-proliferation capacity were not independent (Chi-test value $<10^{-13}$ and $10^{-53}$). Thus, the majority of 7-days-old WT cells with AB or PSG were quiescent.

We then investigated the relationships between AB or PSG formation and mitochondria re-organization. Among cells with vesicular mitochondria, we found that 92% and 65% displayed AB or PSG respectively, about 95% of which being capable to re-proliferate within 6 hr (*Figure 5B*). These observations demonstrate that a cell displaying AB and/or PSG and/or a vesicular mitochondrial network has at least 90% probability of being in a quiescent state after 7 days of culture in YPD. Interestingly, among cells with globular mitochondria, 57% formed AB whereas none assembled PSG (*Figure 5C*). Reciprocally, 5% of the cells with AB displayed globular mitochondria, but none of the cells with PSG displayed globular mitochondria (*Figure 5D*). This strongly suggests that AB can assemble even if cells are unable to proliferate under respiratory conditions, while PSG can not. Accordingly, 99 ± 1% of WT rho zero cells displayed AB (*Figure 5—figure supplement 1A*), but none displayed *bona fide* PSG (*Figure 5—figure supplement 1B*). Thus, AB form in both quiescent and senescent cells while PSG formation occurs only quiescent cells (*Figure 5E*).

## Discussion

In this study, we showed that following carbon source exhaustion in rich medium, yeast proliferation cessation is accompanied by a drastic modification of the mitochondrial network morphology. Indeed, the majority of the cells that will maintain their proliferation capacities for a prolonged period displayed numerous cortical mitochondrial vesicles, while cells that are unable to re-proliferate exhibit globular mitochondria. Mitochondria morphology can thus be utilized as a predictor of cell fate. Using this individual cell property, we found that cell volume, density, and replicative age do not significantly influence the propensity to enter quiescence. We further demonstrated that even if quiescent cells are not identical and display various physical (density, volume), cellular (the presence of AB, PSG, vesicular mitochondria) or historical (replicative age) features, quiescence establishment is most frequently associated with the specific reorganization of cellular machineries.

### Mitochondria reorganization in quiescent cells

Upon quiescence establishment, cells reorganize their tubular mitochondrial network into numerous cortical vesicles. We have shown that this fragmentation can occur even if known components of the fission machinery (Dnm1, Fis1, Caf4 or Mdv1) are absent, or if selective mitophagy is inoperative, attesting that critical actors of the yeast mitochondrial fission apparatus remain to be identified. In fact, the molecular mechanisms underlying mitochondrial reorganization upon quiescence establishment could involve quiescence specific proteins, lipids and/or the modification of cytoplasmic physical properties such as a pH drop or increased viscosity (*Joyner et al., 2016*; *Munder et al., 2016*). Yet, why would small vesicles be more beneficial to quiescent cells than tubules? One hypothesis could be that upon re-entry into proliferation, defective vesicular mitochondria might not be incorporated into the reassembling tubular network, but would rather be targeted for degradation. A vesicular network might thus be a way to easily select and eliminate defective mitochondria. This would not only improve the fitness of the quiescence exit step, but also participate in the rejuvenation of the progeny.

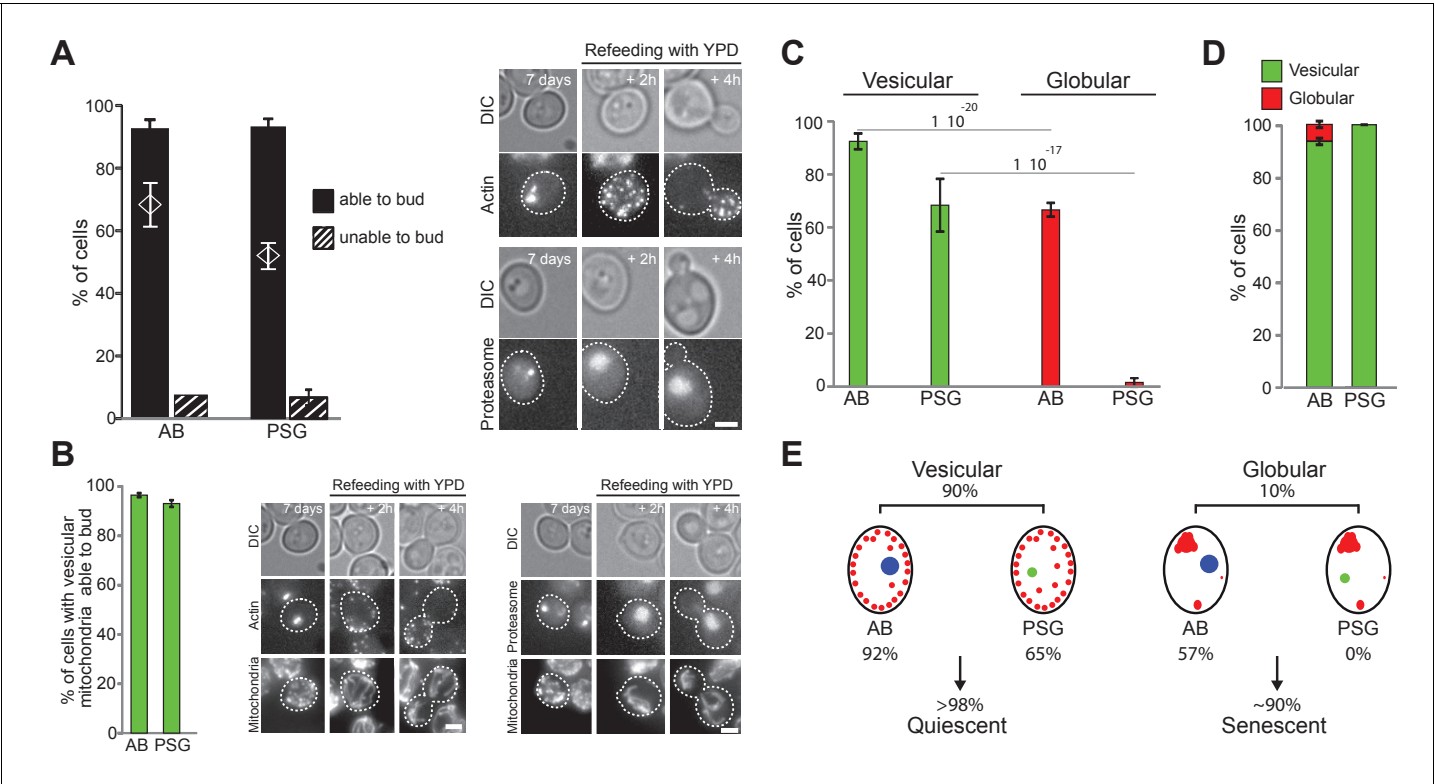

**Figure 5.** Actin, proteasome and mitochondrial network reorganization. WT cells expressing Ilv3-RFP and Abp1-3xGFP (actin) or Scl1-GFP (proteasome) were grown 7 d in YPD and refed on a YPD microscope pad. (**A**) Histograms show the percentage of individual cells displaying AB or PSG able to form a bud within 6 hr after re-feeding. The theoretical 'independence' value (diamond) and representative image series are shown; n > 200, N = 2. (**B**) Percentage of individual cells able to form a bud among cells with both a vesicular mitochondrial network and AB or PSG. Representative image series are shown; n > 300, N = 2. (**C**) Actin and proteasome organization in cells within vesicular and globular mitochondria cell populations; n > 300 N = 2. (**D**) Mitochondria organization in cells with AB or PSG; n > 450 and>300 respectively, N = 2. Bars are 2 μm. (**E**) Cell type and fate distribution among a WT population grown for 7 d. Vesicular mitochondria (small red dots), globular mitochondria (big red dots), AB (big blue dot) and PSG (green dot) are schematized.

DOI: https://doi.org/10.7554/eLife.35685.016

The following source data and figure supplements are available for figure 5:

**Source data 1.** Cell category scoring for each replicated experiment in *Figure 5* panel A, B and D.

DOI: https://doi.org/10.7554/eLife.35685.019

**Figure supplement 1.** Actin and Proteasome localization in WT and rho zero cells, cell volume variation according to mitochondrial network organization.

DOI: https://doi.org/10.7554/eLife.35685.017

**Figure supplement 1—source data 1.** Cell volume measurements in daughter and mother cells depending on their mitochondrial network organization (panel C).

DOI: https://doi.org/10.7554/eLife.35685.018

## Mitochondria and senescence

The definition of senescence is operational and one can never be sure that cells that appear senescent may not re-renter proliferation in very specific environmental conditions. Here, we have shown that cells with globular mitochondria are unable to re-enter the proliferation cycle even 12 hr after re-feeding with glucose. Thus, in our experimental conditions, these cells were considered as senescent.

In *P. anserina*, senescence is accompanied by mitochondrial fragmentation into large globular pieces (*Scheckhuber et al., 2007*) and cristae reticulation (*Brust et al., 2010*). In these cells, the inner membrane forms spherical vesicles, ATP synthase dimers dissociate and large inner/outer membrane contact sites form (*Daum et al., 2013*). In metazoans, abnormal mitochondria, referred to as 'giant mitochondria', have been observed in various senescent models (*Sato and Tauchi,*

*1975*; *Tandler and Hoppel, 1986*; *Murakoshi et al., 1985*) and respiratory chain failure has been proposed to cause premature senescence (*Stöckl et al., 2006*; *Stöckl et al., 2007*). Yet, in these models, it remains unclear if mitochondrial morphological defects contribute to the permanent pro-liferation arrest or if they are rather a consequence of cell entry into senescence (*Ziegler et al., 2015*). Here we establish that in yeast, the formation of globular mitochondria occurs shortly after glucose exhaustion and precludes the loss of re-proliferation capacity, a process that is not synchro-nous within the population.

The loss of re-proliferation capacity is clearly associated with cell's inability to respire. This inca-pacity probably results from an accidental event that either may be the consequence of spontane-ously arising mutation or be indirectly due to a severe metabolic crisis. Why would respiratory deficient cells be more prone to lose their proliferation capacity? Trehalose accumulation is known to be crucial for chronological aging as it fuels re-entry into proliferation (*Samokhvalov et al., 2004*; *Kyryakov et al., 2012*; *Shi and Tu, 2013*; *Cao et al., 2016*; *Laporte et al., 2017*). Respiratory mutants are defective in trehalose stockpiling and immediately mobilize their disaccharide reserves upon glucose exhaustion (*Enjalbert et al., 2000*; *Ocampo et al., 2012*). Limited trehalose storage in respiratory deficient cells with globular mitochondria might thus be responsible for their inability to re-enter the proliferation cycle. In fact, respiratory deficient yeast cells population were previously shown to display a shorter chronological life span than their WT counterparts (*Fabrizio and Longo, 2003*; *Trancíková et al., 2004*; *Bonawitz et al., 2006*; *Aerts et al., 2009*; *Ocampo and Barrientos, 2011*; *Ocampo et al., 2012*). This apparent reduced ability to face age observed at the population scale may result from an increased percentage of senescent cells within the population.

Even if in our experimental conditions, senescence mostly concerns respiratory deficient cells, it is very likely that other individual cell deficiency could lead to senescence in other environmental con-text, for example a starvation for other nutrients. The distinct routes to yeast senescence are likely as yet undiscovered.

## Replicative and chronological aging

During both the replicative and chronological aging processes, the accumulation of damaged macro-molecules is proposed to lead to senescence (*Longo et al., 2012*). Intriguingly, in budding yeast, both aging processes have been associated with mitochondria dysfunction and morphological defects (*Veatch et al., 2009*; *Hughes and Gottschling, 2012*; *Ocampo et al., 2012*; *Fehrmann et al., 2013*; *Beach et al., 2015*; *López-Lluch et al., 2015*; *Dakik and Titorenko, 2016*) and this study). In yeast, chronological age reduces cell's replicative capacity (*Ashrafi et al., 1999*; *Murakami et al., 2012*) as cells that have survived passage through stationary phase for a long period of time have a shortened replicative capacity. Conversely, after the study by Allen *et al*, it was proposed that only daughter cells were capable of entering quiescence (*Allen et al., 2006*). Here we found no difference in the ability to enter quiescence between daughter and mother cells of various replicative age (<15 bud scars) with vesicular mitochondria. Moreover, we found that mother cells of various replicative age (<15 bud scars) and daughter cells have the same propensity to become senescent. Yet, because very old mother cells are rare in a yeast cell population, our indi-vidual cell tracking strategy does not allow us to conclude about replicatively very old cells. There-fore, if chronological age impinges on cell's replicative capacity, the reverse influence is not obvious and, if it exists, it is anecdotal at a population scale because it concerns only very few old mother cells.

## Cell volume and chronological aging

Recently, it was shown that although displaying similar viability, non-dividing small cells lose their proliferative capacity faster than larger cells (*Svenkrtova et al., 2016*). Importantly, these small cells were shown to be unable to respire (*Svenkrtova et al., 2016*). Consistently, we found that very small cells are more prone to display globular mitochondria and rapidly appear senescent. Intriguingly, we also found that cells with globular mitochondria tend to be smaller than cells with vesicular mito-chondria (*Figure 5—figure supplement 1C*). One hypothesis to account for this phenomenon is that since trehalose is a desiccation protectant, cells with globular mitochondria may shrink faster because of a limited trehalose content. Besides, since mother cells with globular mitochondria are larger than daughter cells with vesicular mitochondria (*Figure 5—figure supplement 1C*), we can

exclude the idea that cell volume below a threshold would cause the formation of globular mitochondria.

## Cellular reorganization and quiescence

Quiescence is defined as a reversible arrest of proliferation. In fact, this simple operational concept is not that straightforward. Indeed, while it is easy to understand that quiescence encompasses different molecular processes depending on the cell type, it was demonstrated that for a given cell type, quiescence establishment does not take the same route depending on the nature of the inducing signal (*Coller et al., 2006*; *Klosinska et al., 2011*). In addition, different 'degrees' of quiescence may exist depending on chronological age (*Gookin et al., 2017*; *Coller et al., 2006*; *Laporte et al., 2017*). Thus, quiescence can be viewed as a continuum that tends to senescence and/or ultimately to cell death unless conditions favorable to proliferation are met. Consequently, the idea that quiescence establishment is the result of a universal program is clearly an over-simplification.

This plasticity is further exemplified by the numerous cellular rearrangements that have been associated with proliferation cessation. In yeast, upon carbon source exhaustion, many proteins relocalize to cytosplamic foci (*Brengues et al., 2005*; *Teixeira et al., 2005*; *Sagot et al., 2006*; *Laporte et al., 2008*; *Narayanaswamy et al., 2009*; *Noree et al., 2010*; *Tapia and Morano, 2010*; *Liu et al., 2012*; *Shah et al., 2013*; *Shah et al., 2014*), the actin and microtubule cytoskeletons are re-shaped (*Sagot et al., 2006*; *Laporte et al., 2013*; *Laporte et al., 2015*) and organelles are reorganized (*Yotsuyanagi, 1962*; *Laporte et al., 2013*; *Laporte et al., 2016*; *Laporte and Sagot, 2014*; *Guidi et al., 2015*; *Rutledge et al., 2015* and this study). Some of these rearrangements have been observed in less than 50% of the cells of a non-proliferating population (*Shah et al., 2014*), while others concern more than 95% of individual cells (*Sagot et al., 2006*). Here we show that even if more than 90% of 7 days old cells displaying vesicular mitochondria, or AB or PSG are capable to exit quiescence within few hours, the very few remaining cells are not. Conversely, it is possible to find very few cells that are able to re-proliferate but that do not display these reorganizations. Moreover, the relative proportion of each cell category evolves with chronological age and probably also with the environmental context. Therefore, although statistically significant, none of these rearrangements is an absolute marker of senescent or quiescence. Should such markers exist? If yes, they would have to concern all individuals and should be relevant all along chronological age.

Overall, our data highlight the fact that quiescence does not imply uniformity (*O'Farrell, 2011*) and therefore cannot be apprehended at a population level (*Matson and Cook, 2017*). What, then, makes a quiescent cell? Even if none of the cellular rearrangements identified to date can be used as an absolute quiescence fingerprint, they nevertheless shed light on the obscure biology of quiescent cells. Indeed, while some events occurring upon proliferation cessation are not crucial for cell survival, like telomere clustering (*Laporte et al., 2016*), some of the structures assembled in quiescent cells unambiguously improve the ability to undergo chronological aging. This is the case for actin, microtubule and mitochondria re-organization (*Sagot et al., 2006*; *Laporte et al., 2013* and this study). Deciphering the underlying molecular rationale and the hierarchical relationships between these reorganizations is now critical for a better understanding of quiescence.

## Materials and methods

### Key resources table

| Resource | Designation | Source or reference | Identifiers |
|---|---|---|---|
| Software | ImageJ | Department of Health and Human Services, NIDA | RRID:SCR_003070 |
| Software | GraphPad Prism 5 | GraphPad Software, Inc. La Jolla, USA | RRID:SCR_015807 |

### Yeast strains, growth conditions and individual cell proliferation assay

All the strains used in this study are listed in the *supplementary file 1* and are unless specified, isogenic to BY4741 (*mat a, leu2Δ0, his3Δ0, ura3Δ0, met15Δ0*) or BY4742 (*mat alpha, leu2Δ0, his3Δ0, ura3Δ0, lys2Δ0*), which are available from GE Healthcare Dharmacon Inc. (UK). The W303 and CEN-PK strains are respectively (*mat a, ura3-1, trp1-1, ade2-1, leu2-3,112, his3-11,15, ILV3-RFP::URA3*) and (*mat a; ura3-52; trp1-289; leu2-3,112; his3D1, MAL2-8c, ILV3-RFP::URA3*). BY strains carrying

GFP fusions were obtained from ThermoFisher Scientific (Waltham, MA, USA). The integrative plasmids at the endogenous locus for Ilv3-RFP, Cox4-3xGFP and Atp14-RFP were described previously (*Jimenez et al., 2014*) and for Abp1-3xGFP in *Sagot et al. (2006)*.

Cells were grown in liquid YPD supplemented with adenine or in YPGE in flasks at 30°C, unless specified, as described previously in *Sagot et al. (2006)*. For starvation in nitrogen or phosphorus the media utilized are described in *Klosinska et al. (2011)* and *Pinson et al. (2009)* respectively. For *Figure 1—figure supplement 1A*, cell proliferation was monitored using a Bioscreen Instrument (Oy Growth Curves Ab Ltd, Finland). Cells were inoculated at $OD_{600nm}$=0.01 in 100 µl of YPD or YPGE and grow at 30°C. $OD_{600nm}$ measurements were taken every 10 min.

At a population level, colony forming capacity was addressed by plating 200 cells on YPD, as measured using a Beckman Coulter Multi-sizer 4 (Beckman Coulter Inc., Brea, CA, USA). Each strain was grown in independent culture duplicates and each plating done in triplicate. The capacity of cells to exit quiescence was scored by micro-manipulation of viable cells that is cells not stained with methylene blue (n > 120 cells) at indicated time points, as described in *Laporte et al. (2011)*. Plates were incubated 3 days at 30°C before colony scoring.

For individual cell re-proliferation assays on YPD microscope pads, cells expressing Ilv3-G/RFP were grown in liquid YPD for the time indicated. Cells were then incubated 2–3 min in new YPD and a few µl were spread onto a 2% agarose microscope pad containing YPD. Individual cells were imaged at room temperature in both the DIC and the FP channels, and thus every hour, up to 6 or 12 hr.

## Cell fractionation, cell volume measurement and cell staining

Density gradient fractionation was done by mixing percoll (GE Healthcare Dharmacon Inc., UK) and 1.5 M NaCl in a 9:1 ratio (vol/vol). 1.8 mL of the mix was centrifuged at 21130 g in 2 mL Eppendorf tube for 15 min at 4°C to create the gradient. 500 µL of 7-days-old cells were washed twice in a Tris buffer 0.05 M pH 7, and then placed on the top of the gradient. After 1 hr of centrifugation at 400 g, upper and lower fractions were washed in Tris buffer 0.05 M pH 7 before imaging.

Cell volume measurement were performed either using a Beckman Coulter Multi-sizer 4 (Beckman Coulter Inc., Brea, CA, USA) or manually, using the spheroid formula: $4/3*\pi*a^2*c$, where a and c are the equatorial and polar radius respectively.

Cell viability was scored after 5 min incubation in a solution containing 0.2% of methylene blue (Sigma-Aldrich, Saint Louis, MI, USA) and 2% sodium citrate pH 7. To identify mother and daughter cells, cells were incubated 5 min with Calcofluor white (20 µg/mL, Sigma-Aldrich, Saint Louis, MI, USA), then wash twice in PBS and imaged. To detect mtDNA, cells were fixed 30 min in 4% paraformaldehyde, washed twice with PBS, and incubated 5 min with DAPI (5 µg/mL, Sigma-Aldrich, Saint Louis, MI, USA). Cells were then centrifuged and suspended in PBS with glycerol (70%) and paraphenylenediamine (0.05%). Actin phalloidin staining was performed as described (*Sagot et al., 2006*) with AlexaFluor568 Phalloidin (ThermoFisher Scientific, Waltham, MA, USA).

For the experiments in *Figures 1G–H* and *2* mL of a culture of exponentially growing cells in YPD ($OD_{600nm}$ ~1) were centrifuged, re-suspended in 100 µL YDP, and stained with ConA-FITC (0.2 mg/mL, Sigma-Aldrich, Saint Louis, MI, USA) for 30 min in the dark. In parallel, the remaining culture was centrifuged, filtered, and the medium without cells was used to wash the stained cells three times. Stained cells were then inoculated at $OD_{600nm}$ ~1 in YPD or YPGE.

For *Figure 4F*, unfractionated and dense cell fractions were stained 30 min in the dark at 30°C with ConcA-Alexa647 and ConcA-FITC (0.2 mg/mL, Sigma-Aldrich, Saint Louis, MI, USA), respectively. Cells were washed three times in YPD without glucose, stained with Calcofluor white and imaged on YPD microscope pads.

## Statistical analysis

All the distribution measurements and associated statistics were performed using GraphPad Prism 5 (GraphPad Software, Inc. La Jolla, USA). Unless specified, histograms show the mean and the error bars indicate the SD. To test the 'independence hypothesis' between two parameters, the observed probabilities were noted $P_1$ and $P_2$, the theoretical value expected for two independent parameters being $P_1 \times P_2$. This theoretical value was determined at least on three replicates, allowing mean and SD determination. Chi test values were calculated in Excel (Microsoft Inc, Albuquerque, NM, USA).

## Fluorescence microscopy

Cells were observed on a fully-automated Zeiss 200M inverted microscope (Carl Zeiss, Thornwood, NY, USA) equipped with an MS-2000 stage (Applied Scientific Instrumentation, Eugene, OR, USA), a Lambda LS 300 Watt xenon light source (Sutter, Novato, CA, USA), a $100 \times 1.4$ NA Plan-Apochromat objective, and a five position filter turret. For RFP imaging we used a Cy3 filter (Ex: HQ535/50 – Em: HQ610/75 – BS: Q565lp). For GFP imaging, we used a FITC filter (excitation, HQ487/25; emission, HQ535/40; beam splitter, Q505lp). For calcofluor white imaging we used a DAPI filter (Ex: 360/40 – Em: 460/50 – BS: 400). All the filters are from Chroma Technology Corp. Images were acquired using a CoolSnap HQ camera (Roper Scientific, Tucson, AZ, USA). The microscope, camera, and shutters (Uniblitz, Rochester, NY, USA) were controlled by SlideBook software 5. 0. (Intelligent Imaging Innovations, Denver, CO, USA). For live cell imaging, 2 µL of the cell culture were spotted onto a glass slide and immediately imaged at room temperature.

## Semi-automated method for analysing the mitochondrial network morphology

Data were analyzed using Image J. To define the cell outline, the focal and the $\pm 1$ surrounding planes of DIC images were background subtracted (rolling ball radius: five pixels), projected as maximum projection and re-substracted for the remaining background. To enhance cell outlines, the projection image was filtered using FFT Bandpass filter (1000 pixels for large and three pixels for small structures), resulting in a smoother picture. The $ROIs_{cell}$ were created with the 'analyze particles' feature of image J (size: 0–5000 $pixel^2$; circularity: 0–1; outline display). The area of each $ROI_{cell}$ was measured using an empty frame. Cells 'cut' by the edge of the image or outline fusion between two or more cells were removed by selecting $ROI_{cell}$ between 4.75 and 34 $\mu m^2$. The 'ROI tool' in Image J was used to sequentially measure for each $ROI_{cell}$ the number of objects (NO), the area and the circularity. Circularity was obtained as followed: $(4\pi \times ([Area]/[Perimeter]^2)$ and ranged from 0 (elongated polygon) to 1 (perfect circle).

Four fluorescent subsequent planes were selected around the focal plane. A maximal projection was created and the background subtracted (rolling ball radius: five pixels). The mean ($mean_{Intensity}$) and maximal fluorescence intensities ($max_{Intensity}$) were measured. The $max_{Intensity}$ measured for 7 days old cells was divided by 2.5 and used to threshold the maximal projection ($Mask_{globular}$). The $mean_{Intensity}$ was multiply by 3, and used as a threshold to generate the $Mask_{initial}$. Using the 'analyze particles' tool of image J, we generated the $Mask_{tubule}$ by filtering the $Mask_{initial}$ with a size of 20–5000 $pixel^2$ and a circularity of 0–0.5 (masks display). The $Mask_{vesicules}$ was created with the same approach but using a size of 0–12 $pixel^2$ and a circularity between 0.85–1. Finally, using the 'Image calculator' feature of image J, we generated the $Mask_{fragment} = Mask_{initial} - (Mask_{tubule} + Mask_{vesicules})$.

$ROIs_{cell}$ were applied to the $Mask_{globular}$. A $ROI_{cell}$ in which the signal area >0.0018 $\mu m^2$ (*i.e.* >2 pixels) was considered as a cell with globular mitochondria. $ROIs_{cell}$ were applied to $Mask_{initial}$, $Mask_{tubule}$, $Mask_{vesicules}$ and $Mask_{fragment}$. In each $ROI_{cell}$, the numbers of objects ($NO_{\_Mask}$) was determined. In case of $NO_{\_initial}$ >4, the percentage of object in each mask was calculated: % $NO_{\_tubule} = NO_{\_tubule}/NO_{\_initial}$, % $NO_{\_vesicle} = NO_{\_vesicle}/NO_{\_initial}$ and % $NO_{\_fragment} = NO_{\_fragment}/NO_{\_initial}$. For each cell, when the % $NO_{\_vesicules}$ was the highest, the cell was considered as 'vesicular' as it displays an important number of objects with high circularity.

'Tubular' and 'fragmented' mitochondria were distinguished based on two parameters: (1) the axial property of tubular network (low circularity) and (2) their relative areas (*i.e.* the total areas/number of objet). The % $NO_{\_}$ and the (area/$NO_{\_}$) for each $ROI_{cell}$ on $Mask_{tubule}$ and $Mask_{fragment}$ were compared. If for a $Mask_{tubule}$ these two parameters were higher than for the $Mask_{fragment}$, the cell was defined as 'tubular' if these two parameters were lower, the cell was defined as 'fragmented'. If none of these criteria were met, a new filter was applied. A cell was considered as 'vesicular' if $NO_{\_vesicules} = NO_{\_initiale}$. A cell was considered as 'tubular' if $Area_{tubule}/NO_{\_tubule} > Area_{vesicules}/NO_{\_vesicules}$ and > $Area_{fragment}/NO_{\_fragment}$. Finally a cell was considered as 'fragmented' if $Area_{fragment}/NO_{\_fragment} > Area_{vesicules}/NO_{\_vesicules}$ and > $Area_{tubule}/NO_{\_tubule}$. These filters were used for $NO_{\_initial}$ <4.

## Miscellaneous

Glucose concentration was measured using the d-Glucose/d-Fructose UV test kit (Roche, SW). The Ethidium Bromide is from Sigma-Aldrich (Saint Louis, MI, USA).

## Acknowledgements

We express our profound gratitude to B Daignan-Fornier, J-P Javerzat, A Devin, and A Mourrier, for precious discussions about our work. We thank B Daignan-Fornier and D McCusker for comments on this manuscript. We are grateful to A Devin, J-L Parrou and F Doignon for providing reagents. This work was supported by the Agence Nationale pour la Recherche (ANR-12-BSV6-0001), the University of Bordeaux and the Centre National de la Recherche Scientifique.

## Additional information

### Funding

| Funder | Grant reference number | Author |
| --- | --- | --- |
| Agence Nationale de la Recherche | ANR-12-BSV6-0001 | Damien Laporte<br>Isabelle Sagot |
| Centre National de la Recherche Scientifique | | Damien Laporte<br>Laëtitia Gouleme<br>Isabelle Sagot |
| Université de Bordeaux | | Damien Laporte<br>Laëtitia Gouleme<br>Laure Jimenez<br>Ines Khemiri<br>Isabelle Sagot |

The funders had no role in study design, data collection and interpretation, or the decision to submit the work for publication.

### Author contributions

Damien Laporte, Conceptualization, Resources, Data curation, Formal analysis, Supervision, Validation, Investigation, Visualization, Methodology, Writing—original draft, Writing—review and editing; Laëtitia Gouleme, Laure Jimenez, Ines Khemiri, Investigation, Experimental work; Isabelle Sagot, Conceptualization, Data curation, Formal analysis, Supervision, Funding acquisition, Validation, Investigation, Visualization, Methodology, Writing—original draft, Project administration, Writing— review and editing

### Author ORCIDs

Isabelle Sagot http://orcid.org/0000-0003-2158-1783

### Decision letter and Author response

Decision letter https://doi.org/10.7554/eLife.35685.029
Author response https://doi.org/10.7554/eLife.35685.030

## Additional files

### Supplementary files

• Supplementary file 1. Table with the genotype of the strains used in this study.
DOI: https://doi.org/10.7554/eLife.35685.020

• Transparent reporting form
DOI: https://doi.org/10.7554/eLife.35685.021

All data analysed during this study are included in the manuscript. Due to their large size, raw microscope images acquired for this study can be made available upon request.

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
