## [Decision Letter]

Thank you for submitting your article "Mitochondria reorganization upon proliferation arrest predicts individual yeast cell fate" for consideration by *eLife*. Your article has been reviewed by three peer reviewers, and the evaluation has been overseen by and Randy Schekman as the Senior/Reviewing Editor. The reviewers have opted to remain anonymous.

The reviewers have discussed the reviews with one another and the Reviewing Editor has drafted this decision to help you prepare a revised submission.

Summary:

In the manuscript, the authors tackle the difficult question of identifying new ways to distinguish between quiescent and senescent cells in a yeast population. Several past studies identified methods to do this, but to date, all have some potential limitations. In this study, the authors show that as yeast cells enter stationary phase, mitochondrial structure changes from a tubular network to a collection of small fragmented vesicles in the majority of cells. However, a subpopulation of cells, ~10%, contain large globular mitochondria. The authors track the fate of these non-dividing cells, and show that ~90% of cells with globular mitochondria are senescent, while only 10% of cells with vesicular mitochondria are senescent. Based on this observation, the authors suggest that tracking mitochondrial morphology is a mechanism by which to identify quiescent and senescent cells in a population. They then go on to show that previously identified phenotypes such as replicative age, cell density, and cell volume do not always correlate or impact a cell's ability to undergo senescence or quiescence. Overall, this is a high-quality manuscript that will have a significant impact on the field of quiescence. The data are clear, and the manuscript is well constructed.

Essential revisions:

1) A major weakness is that detailed methods are not provided for how the mitochondrial network is characterized. A fragmented network seems quite similar to a globular network. Use of statistical methods is not provided for how these two network morphologies are characterized. Have the authors developed an algorithm to sort the different morphologies and what are the parameters that are considered (i.e., length vs. width, etc.). If the morphology is characterized by eye, separating fragmented from vesicular seems like this could vary by user. This seems critical to include in the Materials and methods as other researchers may want to use this approach.

2) The authors have clearly demonstrated that cells with globular mitochondria are more likely to enter senescence, while those with fragmented vesicular mitochondria are more likely quiescent. This is interesting and suggests a link to mitochondrial dysfunction and senescence. However, I am concerned that the mitochondrial phenotypes do not provide a great way to identify senescent versus quiescent cells in a population as the authors suggest. Currently, the way the data is presented in Figure 2, including the summary in 2G, it appears that identifying globular mitochondria containing cells is a great way to separate out quiescent/senescent. However, when analyzing a total cell population, fragmented mitochondrial cells make up ~90% of the population, and ~10% of those are senescent. Globular mitochondrial cells are ~10% of the population, and ~90% are senescent. That means that if one looks at 100 cells, about 18 cells would be senescent, and the proportion that are globular and fragmented would be ~50:50. Thus, one can only identify about half of the cells that are senescent in the population using mitochondrial morphology. The authors should include a representation of the Figure 2B data in this form, showing what fraction of the total senescent cells in the population have fragmented or globular mitochondria. They should also update Figure 2G to provide a more accurate depiction of this. As it stands, 2G does not show that about 50% of the senescent cells contain vesicular mitochondria. The bigger issue is that when presented this way, it becomes apparent that while mitochondrial phenotype correlates with probability of senescence, it can only identify a subpopulation of senescent cells in the population. The authors should make sure their Discussion highlights this in the manuscript.

3) Another concern in Figure 2B is that the authors only examine the cells for 6 hours. This doesn't seem long enough to conclude that these cells will not divide again. Is it possible that globular mitochondrial containing cells take longer to exit quiescence? The authors should include an experiment where they track cells for much longer than 6 hours, perhaps using their ConA-FITC labeling technique.

4) A weakness in this manuscript is that the authors propose that the ability to respire is important for moving into quiescence; testing the distribution on YPGE would be useful for clarifying this aspect further as one would not expect cells on YPGE to become senescent. The authors also tend to make larger claims in the Discussion that have not been tested such as the proposal that vesicular mitochondria may be able to be eliminated more easily; testing this with an autophagy assay is necessary to confirm this mechanism.

5) Subsection “Mitochondrial network reorganization upon proliferation cessation”, first paragraph: Since the BY deletion-strain collection is prone to accumulate secondary mutations these results should be confirmed with hand-made deletion strains or deletions in other strain backgrounds used in this study (W303 or CEN-PK).

[Editors' note: further revisions were requested prior to acceptance, as described below.]

Thank you for resubmitting your work entitled "Mitochondria reorganization upon proliferation arrest predicts individual yeast cell fate" for further consideration at *eLife*. Your revised article has been favorably evaluated by Randy Schekman (Senior/Reviewing Editor) and two expert reviewers.

The manuscript has been improved but there is one remaining issue that need to be addressed before acceptance, as outlined below. Please consider the remark of reviewer #1 and respond in writing or with additional analysis to address the possibility of an acquired suppressor mutation.

I will evaluate your response and issue a decision in consultation with reviewer #1.

*Reviewer #1:*

The authors have carefully addressed all previous concerns raised by the reviewers. The manuscript and data presented are high quality and will be of significant interest to the readers. A minor concern stemming from new data included in the manuscript is the phenotype of the *mdm34Δ* ERMES mutant, which the authors find is not respiratory deficient and doesn't share similar morphology to the *mmm1Δ* mutant. This is in disagreement with previous reports that *mdm34* deletion strains are respiratory deficient. I wonder if it's possible that the *mdm34* mutant cells the authors are working with have acquired a suppressor mutation, perhaps in *Vps13*, which is very common with these mutants (see PMID: 26370498). The authors may want to sequence the *Vps13* locus in their *mdm34* mutant strains, and consider excluding this data from the manuscript if the *mdm34Δ* strain contains a suppressor mutation that renders it respiration positive.

Reviewer #2:

Good revision.

---

## [Author Response]

Essential revisions:1) A major weakness is that detailed methods are not provided for how the mitochondrial network is characterized. A fragmented network seems quite similar to a globular network. Use of statistical methods is not provided for how these two network morphologies are characterized. Have the authors developed an algorithm to sort the different morphologies and what are the parameters that are considered (i.e., length vs. width, etc.). If the morphology is characterized by eye, separating fragmented from vesicular seems like this could vary by user. This seems critical to include in the Materials and methods as other researchers may want to use this approach.

The mitochondrial network starts to fragment shortly after the diauxic shift (Figure 1A). With time, the mitochondrial fragments progressively appear shorter and shorter. We believe that the vesicular network observed in about 85% of the cells at 7 days is an ultimate form of fragmentation of the tubules observed in proliferating cells. Therefore, we think that the physiological differences between cells with vesicular mitochondria and cells with short mitochondrial tubules are loose. This is now clearly stated in the text.

Yet, we have developed a semi-automated method to discriminate the different cell categories. Briefly, cells with globular mitochondria were identified based on fluorescence intensity and cells with vesicular or fragmented mitochondria were differentiated using object number per cell and object circularity. This method is provided in the Materials and methods section. Author response image 1 presents the results obtained using the above method in the most complex situation, i.e. the kinetic of entry of a WT cell population into stationary phase. This figure is very similar to Figure 1A.

Importantly, cells with globular mitochondria are very easy to distinguish from cells with vesicular mitochondria just by eye or by using our semi-automated method. That is why we are confident in our scoring of “vesicular” versus “globular”; categories that are the most important for our study.

2) The authors have clearly demonstrated that cells with globular mitochondria are more likely to enter senescence, while those with fragmented vesicular mitochondria are more likely quiescent. This is interesting and suggests a link to mitochondrial dysfunction and senescence. However, I am concerned that the mitochondrial phenotypes do not provide a great way to identify senescent versus quiescent cells in a population as the authors suggest. Currently, the way the data is presented in Figure 2, including the summary in 2G, it appears that identifying globular mitochondria containing cells is a great way to separate out quiescent/senescent. However, when analyzing a total cell population, fragmented mitochondrial cells make up ~90% of the population, and ~10% of those are senescent. Globular mitochondrial cells are ~10% of the population, and ~90% are senescent. That means that if one looks at 100 cells, about 18 cells would be senescent, and the proportion that are globular and fragmented would be ~50:50. Thus, one can only identify about half of the cells that are senescent in the population using mitochondrial morphology. The authors should include a representation of the Figure 2B data in this form, showing what fraction of the total senescent cells in the population have fragmented or globular mitochondria. They should also update Figure 2G to provide a more accurate depiction of this. As it stands, 2G does not show that about 50% of the senescent cells contain vesicular mitochondria. The bigger issue is that when presented this way, it becomes apparent that while mitochondrial phenotype correlates with probability of senescence, it can only identify a subpopulation of senescent cells in the population. The authors should make sure their Discussion highlights this in the manuscript.

To represent our data we have two possibilities: either by “mitochondrial morphology”, like we did in Figure 2H (the former Figure 2G), or by “cell fate” as proposed by the reviewers. The problem with the representation by “cell fate” is that the percentages greatly vary with the chronological age of the culture. Indeed, as shown Figure 2C (that is now 2D), after 2 days of culture, about 70% of the cells with globular mitochondria are able to re-enter the proliferation cycle. However, after 7 days of culture, 90% of the cells with globular mitochondria are unable to re-enter the proliferation cycle. Thus if we consider 10 globular cells, after 2 days of culture, 3 will be senescent while after 7 days of culture, 9 will be senescent. By contrast, the representation by “morphology” is independent of the culture age (at least for the first 5 weeks) since as soon as the diauxic shift is passed, the percentage of cells with globular or vesicular mitochondria stay about constant (Figure 1—figure supplement 1B).

Importantly, our objective is not to identify *all* the senescent cells. Indeed, this exercise is always open to criticisms since a cell is considered as senescent until a way to make it re-enter proliferation is found. In the new Figure 2C, as suggested by the reviewers in the point #3 below, we show that if we leave the cells longer time to re-proliferate after re-feeding on pad, almost all cells with vesicular mitochondria will re-proliferate (98% +/-2 after 12h compared to 90% after 6h on the pad). By contrast, whatever the time on the pad, 90% of the 7 days old cells with globular mitochondria do not re-enter the proliferation cycle and can thus be considered as senescent.

For all these reasons, we believe that our representation by “mitochondrial morphology” is less misleading than a representation by “cell fate”, which is strongly influenced by both the time in culture and the time allowed to re-proliferate. Yet, we have softened Figure 2G by adding “apparently senescent”, and discussed that point more clearly in the Discussion section.

3) Another concern in Figure 2B is that the authors only examine the cells for 6 hours. This doesn't seem long enough to conclude that these cells will not divide again. Is it possible that globular mitochondrial containing cells take longer to exit quiescence? The authors should include an experiment where they track cells for much longer than 6 hours, perhaps using their ConA-FITC labeling technique.

As explained in the manuscript, because most cells rapidly re-proliferate on the pad and invade the field of analysis, it is not possible to track individually a large number of cells for more than 6h after re-feeding. However, as suggested by the reviewer, we have left more time for cells to re-proliferate on pads that were inoculated with very few cells. In those experiments, individual cell full progeny cannot be tracked because of overcrowding (see new Figure 2C).

Nevertheless, we observed that 98% +/-2 of 7 days old cells with vesicular mitochondria re-proliferate after 12h on the pad (compare to 90% after 6h). Thus most of the 7d-old cells with vesicular mitochondria were quiescent. By contrast, and importantly, increasing the time on the pad did not change cells with globular mitochondria ability to re-enter proliferation (13% +/- 10% after 6h and 9%+/- 1% after 12h). These data are now presented in the new Figure 2C.

4) A weakness in this manuscript is that the authors propose that the ability to respire is important for moving into quiescence; testing the distribution on YPGE would be useful for clarifying this aspect further as one would not expect cells on YPGE to become senescent.

As suggested by the reviewer, we have addressed the morphology of the mitochondrial network of cells grown in YPGE or in rich medium containing lactate and as expected by the reviewer, we found no cells with globular mitochondria. Importantly, almost all the cells grown for 4 or 9 days in YPGE or lactate were viable and capable to give rise to a progeny. These data are shown in the new Figure1—figure supplement 1F.

The authors also tend to make larger claims in the Discussion that have not been tested such as the proposal that vesicular mitochondria may be able to be eliminated more easily; testing this with an autophagy assay is necessary to confirm this mechanism.

This is just a hypothesis proposed in the Discussion section. In fact, if true, quiescent cells should have two distinct types of mitochondrial vesicles: some functional and some non-functional. We are currently testing this possibility and some preliminary data are provided as “confidential data for the reviewers” but we believe that testing this hypothesis and deciphering the underlying mechanism will involve many complex experiments that are beyond the scope of this manuscript.

5) Subsection “Mitochondrial network reorganization upon proliferation cessation”, first paragraph: Since the BY deletion-strain collection is prone to accumulate secondary mutations these results should be confirmed with hand-made deletion strains or deletions in other strain backgrounds used in this study (W303 or CEN-PK).

We are aware that these strains could be unstable. That is why the experiments of Figure 1B were made with deletion strains obtained in the lab (not from a library), with cells grown form freshly dissected spores (obtained by crossing with WT) and repeated several times with at least three different clones.

As shown in Author response image 2 and as expected from the phenotype described in literature for proliferating *dnm1Δ, fis1Δ, mdv1Δ and caf4Δ*, the strains we have used in our study display a highly interconnected mitochondrial network when they actively proliferate.

**Author response image 2. respfig2:** 

Further, after 7 days of growth in YPD, when these mutant cells were re-fed, they rapidly re-form the expected interconnected mitochondrial tubules (as shown for *dnm1Δ* in Author response image 3). Therefore, the strains we are using are indeed impaired for mitochondria fission.

**Author response image 3. respfig3:** 

In addition, during the time allowed for the revision, we were able to construct and backcross *atg32Δ* and *fis1Δ* deletions in the W303 background. The mitochondria organization observed in these cells after 7 days of growth in YPD was similar to the one observed for cells of the BY background presented in Figure 1B.

**Author response image 4. respfig4:** 

Because these data are only confirmatory and just prove that our strains are OK, we chose not to add them in the revised manuscript but if the reviewers believes we should, we will be happy to do so.

[Editors' note: further revisions were requested prior to acceptance, as described below.]

Reviewer #1:The authors have carefully addressed all previous concerns raised by the reviewers. The manuscript and data presented are high quality and will be of significant interest to the readers. A minor concern stemming from new data included in the manuscript is the phenotype of the mdm34Δ ERMES mutant, which the authors find is not respiratory deficient and doesn't share similar morphology to the mmm1Δ mutant. This is in disagreement with previous reports that mdm34 deletion strains are respiratory deficient. I wonder if it's possible that the mdm34 mutant cells the authors are working with have acquired a suppressor mutation, perhaps in Vps13, which is very common with these mutants (see PMID: 26370498). The authors may want to sequence the Vps13 locus in their mdm34 mutant strains, and consider excluding this data from the manuscript if the mdm34Δ strain contains a suppressor mutation that renders it respiration positive.

In our hands, at the individual cell level, only 16% of the *mdm34Δ* cells were respiratory defective. The *mdm34Δ* mutant was originally described as unable to grow on respiratory carbon sources (Dimmer et al., 2002; Youngman et al., 2004) but Youngman et al. described that *mdm34Δ* strains rapidly “acquire” the ability to grow on non-fermentable carbon source.

Lang et al. (2015) have demonstrated that two dominant mutations in the *VPS13* gene can suppress the *mdm34Δ* respiration defective phenotype (Lang et al., JCB, 2015). These mutations are at position 1814: TTA⋄TCA (L605S) and at position 3665: CCT⋄CAT (P1222H). As requested by the reviewer, we have sequenced the *VPS13* allele in the *mdm34Δ* Ilv3-RFP strain used in Figure2—figure supplement 1E-F and found none of the above mentioned mutations (please see the chromatograms in Author response image 5).

**Author response image 5. respfig5:** 

In addition, we have backcrossed our *mdm34Δ* Ilv3-RFP strain with a WT strain and found a “slow growth on glucose medium” phenotype that segregates 3:1, suggesting that indeed, a single mutation suppresses the previously reported “slow growth on glucose medium” phenotype of the *mdm34Δ* mutation.

In our strain, the *mdm34* gene is disrupted with the KanMX gene (Research Genetics strain from Invitrogen). Accordingly, the colonies with a “slow growth on glucose medium” phenotype are resistant to kanamycine (please see Author response image 6). Furthermore, as expected if a single mutation suppresses the *mdm34Δ* “slow growth on glucose medium” phenotype, the other [KanR] colony in each tetrad grew normally on YPD. Moreover, after 24h, the [*mdm34Δ “slow growth on glucose medium”; KanR]* colonies did not grow on YPGE, while, as expected, the *[mdm34Δ+suppressor; KanR]* colonies did. This demonstrate that a single mutation suppresses the respiratory deficient phenotype of the *mdm34Δ* Ilv3-RFP strain used in Figure2—figure supplement 1E-F.

Yet, importantly, after 2 days, many of the newly obtained *mdm34Δ* colonies that were undetectable on YPGE 24h after replica-plating of the dissection plates, did grow, although slowly. This indicates that the newly obtained *mdm34Δ* strains have acquired suppressor that provide them the ability to respire, just as described previously Yougman et al., 2004.

**Author response image 6. respfig6:** 

Altogether, this indicates that the *mdm34Δ* Ilv3-RFP strain used in Figure2—figure supplement 1E-F contains a mutation that suppresses its respiratory deficient phenotype. Because we do not know the nature of this mutation and as the above experiments indicate that the *mdm34Δ* strain is highly unstable, we chose to remove the data concerning *mdm34Δ* in Figure2—figure supplement 1 and the corresponding few lines of text in our manuscript.